# Thermo-Electrical Conduction of the 2,7-Di([1,1′-Biphenyl]-4-yl)-9H-Fluorene Molecular System: Coupling between Benzene Rings and Stereoelectronic Effects

**DOI:** 10.3390/molecules25143215

**Published:** 2020-07-14

**Authors:** Judith Helena Ojeda Silva, Juan Sebastián Paez Barbosa, Carlos Alberto Duque Echeverri

**Affiliations:** 1 Grupo de Física de Materiales, Facultad de Ciencias, Universidad Pedagógica y Tecnológica de Colombia, Tunja 150003, Colombia; juansebastian.paez@uptc.edu.co; 2 Laboratorio de Química Teórica y Computacional, Grupo de Investigación Química-Física Molecular y Modelamiento Computacional (QUIMOL), Facultad de Ciencias, Universidad Pedagógica y Tecnológica de Colombia, Tunja 150003, Colombia; 3 Grupo de Materia Condensada-UdeA, Instituto de Física, Facultad de Ciencias Exactas y Naturales, Universidad de Antioquia UdeA, Calle 70 No. 52-21, Medellín 050030, Colombia; cduque_echeverri@yahoo.es

**Keywords:** green’s functions, decimation process, thermal and electric properties, stereoelectronic effect

## Abstract

Theoretical and analytical thermal and electrical properties are studied through the 2,7-Di([1,1′-biphenyl]-4-yl)-9H-fluorene aromatic system as a prototype of a molecular switch. Variations of the dihedral angles between the two Benzene rings at each end of the molecule have been considered, thus determining the dependence on the structural variation of the molecule when the aromatic system is connected between metal contacts. The molecule is modeled through a Tight-Binding Hamiltonian where—from the analytical process of decimation and using Green’s functions—the probability of transmission (*T*) is calculated by using the Fisher–Lee relationship. Consequently, the thermal and electrical transport properties such as I−V curves, quantum noise (*S*), Fano factor (*F*), electrical conductance (*G*), thermal conductance (κ), Seebeck coefficient (*Q*), and merit number (ZT) are calculated. The available results offer the possibility of designing molecular devices, where the change in conductance or current induced by a stereoelectronic effect on the molecular junctions (within the aromatic system) can produce changes on the insulating–conductive states.

## 1. Introduction

Given the current rate of technological advancement, the smallest components that begin to be functional such as individual molecules and small groups of molecules (carbon nanotubes, metal wires, or nanoscale semiconductors) are taking great steps to be used in molecular electronics and consequently contribute to the development of nanoscience. The first theoretical and experimental efforts in this field began in the 1970s with the work of Aviram and Ratner who described how a modified salt can work as a traditional diode in an electrical circuit [1]. Although it did not have much impact at that time, this represented an important first attempt to predict and characterize an electronic phenomenon on a molecular scale: the use of cellular automata, the invention of the Scanning Tunneling Microscope (STM) and its corresponding application for the study of the electronic transport properties of individual molecules using the Mechanically Controllable Break Junctions (MCBJ), reconfigurable switches, use of organic molecules that exhibit rectifying behavior or a very pronounced negative differential resistance, and gate electrodes in single-molecule junctions, among others [2,3,4,5,6,7,8,9,10,11,12].

Molecular systems based on benzene rings have become important devices that can be used in molecular electronics. They have been characterized with superconducting properties as well as with special characteristics to be used in solar cells, nanotransistors, organic light-emitting diode (OLED), to name a few [13,14]. In particular, the 2,7-Di([1,1′-biphenyl]-4-yl)-9H-fluorene molecule, which is based on benzene rings, is an organic molecule with high potential for various future applications. It is one of the most promising candidates to act as a Light Emitting Diode (LED); its nanofibers have properties as waveguides and could serve as future nanoscale optical connectors on solid surfaces [15]. However, a new class of aromatic ring molecular nanotransistors based on single-molecule graphene-molecule-graphene junctions was recently reported through the use of an ionic liquid gate and using the hexaphenyl, biphenyl, and terfeline molecules (the hexaphenyl molecule has a similar structure to the 2,7-Di([1,1′-biphenyl]-4-yl)-9H-fluorene molecule). In addition, it was found that graphene-hexaphenyl-graphene molecular bonds are stable for a long time at room temperature; these studies open the way to manufacture stable single-molecule devices at room temperature [16,17,18,19].

Over the past few years, researchers in chemistry, physics, and nanotechnology have begun to understand stereoelectronics adequately. This is the means through which it is possible to interpret how and why the electronic properties of molecules vary and can be modified in the process of shaping physical–chemical systems. With the appearance of single-molecule techniques, the necessary instruments and technology have been created to follow the conformation of molecules step by step. By introducing the stereoscopic effect, for example, in the aromatic structure of biphenyl, it is possible to modify the structural and physical properties of an organic functional molecule. Aiming to investigate the process of intramolecular conformation of biphenyl and how this affects the dynamics of its transport properties at the level of an isolated molecule, Xin et al. [19] designed an aromatic hexaphenyl chain molecule. The medial biphenyl was placed using a form of fluorene in order to fix the dihedral angle. Additionally, it has been covalently sandwiched between the contacts of a graphene dot, and thus, stable single-molecule bonds have been created. The authors report—mainly at a temperature of 120 K—a stochastic change between two different states of conductance in the I-V curves, and have demonstrated that they remain in a state of low conductance up to 140 K. Biphenyl, as the elementary unit of organic functional materials, has been shown to have significant potential for use in organic field-effect transistors, organic light-emitting diodes, and solar cells. Both experimental and theoretical researchers suggest that the stereoelectronic effect of the biphenyl-based aromatic structures can significantly affect the structural and photophysical properties of organic functional molecules [20,21,22,23,24,25]. Li et al. [26] investigated stereoelectronic effects in a hexaphenyl aromatic chain. In their work, the central biphenyl was introduced in a fluorene form to fix the dihedral angle. Following the same approach, and in order to analyze the single-molecule charge transport properties, the authors have included an amine-terminated hexaphenyl aromatic chain located between carboxylic-acid-terminated graphene electrodes.

In this work, we consider the theoretical and analytical study of thermal and electrical properties through the aromatic 2,7-Di([1,1′-biphenyl]-4-yl)-9H-fluorene system as a prototype of a molecular switch wire, by including the dependence on the dihedral angles between the two external benzene rings at each end and the interactions between the molecule and the contacts (see Figure 1). Such effects of the structural variation of the molecule are analyzed when the aromatic system is connected between metal contacts and are modeled through a Tight-Binding Hamiltonian (TBH), taking into account an analytical process of decimation by using the Green’s functions to calculate the probability of transmission [27,28,29,30,31] and the quantum fluctuations in the transport properties—which are known as the noise power spectrum. The last one, in a steady state, is described by the shot noise. The noise power provides important information about the electronic correlation by means of the Fano factor (F), which indicates whether the magnitude of the noise reaches Poisson (F=1) or sub-Poisson (F<1) limits [27,32,33]. In addition to the above, using Landauer-Büttiker formalism, we calculate the electrical and thermal transport properties (such as conductance, thermal conductance, I-V curves, and Seebeck coefficient) in order to characterize the system.

The renormalization method is a method that transforms the real space into an effective space that generates the systematic reduction of the Hamiltonian dimension, i.e., the set of linear equations that characterize the system becomes a set of nonlinear equations with effective energies and couplings with fewer degrees of freedom. The renormalization method is a highly efficient and very low computational cost (compared to ab initio, such as density functional theory (DFT) methods). For this reason, to calculate the transport properties in this work, the renormalization process is done under the inclusion of Feynman paths using Green’s functions, which can be considered as propagators (Dyson’s equation) that carry the charge transport information through the molecular system and provide an alternative in the solution of the Schrödinger equation in a numerical and analytically stable way. It is important to note that the renormalization method used in this work has been implemented in the calculation of quantum transport properties through quantum wire systems, aromatic molecular systems, DNA molecules, among others, and have been compared with transport properties through similar systems, determined by DFT calculations and/or experimental results [28,29,30,31,34].

The manuscript is organized as follows: In Section 2 the molecular system model, which is based in a TBH, is introduced. In Section 3, the methodology is described. Section 4 is devoted to the results and corresponding discussions. Finally, the conclusions are given in Section 5.

## 2. The Molecular System Model

To describe the molecular system as an electronic device, which is connected to two metal contacts (See Figure 1), we introduce the following Hamiltonian in the Tight-Binding approach with first neighbors:(1)$$H=H_{0}+H_{c}+H_{I}\;,$$
where $H_{0}$ corresponds to the molecular system Hamiltonian, which is given by
(2)$$H_{0}=\sum_{n}E_{n}a_{n}^{+}a_{n}+\sum_{n}t_{n}\left(a_{n}^{+}a_{n+1}+a_{n+1}^{+}a_{n}\right)\;,$$
where $H_{C}$ represents the Hamiltonian of the metal contact, and $H_{I}$ is related to their interaction with the molecular system; they are given by
(3)$$H_{c}=\sum_{k_{L}}\varepsilon_{k_{L}}d_{k_{L}}^{+}d_{k_{L}}+\sum_{k_{R}}\varepsilon_{k_{R}}d_{k_{R}}^{+}d_{k_{R}}$$
and
(4)$$H_{I}=\sum_{k_{L}}\Gamma_{L}d_{k_{L}}^{+}c_{L}+\sum_{k_{R}}\Gamma_{R}d_{k_{R}}^{+}c_{R}+h.c.\;\;.$$

Here, $E_{n}$ ($=E_{c},\;E_{h}$) are the energy sites corresponding to the carbon or hydrogen atoms, respectively, $t_{n}$ is the coupling between carbon atoms, $t_{n}=t_{\vartheta },\;t_{\delta }$ where $t_{\vartheta }$ is the coupling between the two extreme benzene rings of the molecule (dihedral angle) and $t_{\delta }$ are the C−C or C=C bonds. In addition, $a_{n}^{+}$ and $a_{n}$ are the creation and destruction operators, where a state is created or destroyed on the site *n*, respectively. Moreover, $d_{k_{L(R)}}$ is the operator which creates an electron in the $k_{L(R)}$ state, with $\epsilon_{k_{L(R)}}$ energy, while $\Gamma_{L,R}$ is the coupling between each contact (left and right) with extreme atom site on the left or right sides of the molecular system.

## 3. Methodology

Using the Landauer formalism, we study the fluctuations in both current and temperature through molecular wire composed of benzene rings connected to two semi-infinite leads [27,32,33]. The calculations are based on Green’s function techniques within a real-space renormalization approach for molecular systems. This allows us to reduce the many-body problem into a multichannel scattering problem for a single electron. The Green’s function of the molecular systems coupled to the contacts is related to the tight-binding Hamiltonian through the expression G=1/(z−H), where *z* is a complex term given by z=E−iη and η is a infinitesimal term. Therefore, *G* can be calculated by using the Dyson equation, given by
(5)$$G=G_{0}+G_{0}\left(\Sigma_{L}+\Sigma_{R}\right)G\;,$$
where $G_{0}$ is the bare Green’s function of the isolated molecular system and $\Sigma_{L}$ and $\Sigma_{R}$ are the self-energies of the left and right contact, respectively. The transmission probability can be obtained by using the Fischer–Lee relationship [32]:(6)$$T\left(E\right)=Tr\left[\Gamma_{L}G^{r}\Gamma_{R}G^{a}\right]\;,$$
being $\Gamma_{L(R)}=i\left(\Sigma_{L(R)}-\Sigma_{L(R)}^{†}\right)$ the spectral matrix density of the left (right) contact. Here, we have used $\Sigma_{L}=\Sigma_{R}=-i\Gamma /2$.

Due to the high complexity while describing the 2,7-Di([1,1′-biphenyl]-4-yl)-9H-fluorene molecular structure by the Hamiltonian, its solution is rather complicated. Therefore, in order to numerically solve the transport properties on this system, we perform the Decimation process as a valid approximation. It is based on considering the molecular system into an effective linear chain, such as that shown the Figure 2, obtaining the respective Green’s functions $G_{0}$ that contain all the information of the planar molecule.

Then, the transmission probability for the new one-dimensional effective molecular system can be written as
(7)$$T\left(E\right)=\Gamma_{L}\Gamma_{R}{\left|G_{1N}\right|}^{2}\;,$$
with
(8)$$G_{1N}=\frac{G_{1N}^{0}}{\left(1-\Sigma_{L}G_{11}^{0}\right)\left(1-\Sigma_{R}G_{NN}^{0}\right)-\Sigma_{L}\Sigma_{R}{\left(G_{1N}\right)}^{2}}\;.$$

The $G_{1N}^{0}$, $G_{NN}^{0}$, and $G_{11}^{0}$ Green’s functions can be analytically determined by using renormalization techniques (the details of the Green’s function calculations are presented in Appendix A). With $G_{NN}^{0}=G_{11}^{0}$ and $\Gamma_{L}=\Gamma_{R}=\Gamma$, we can rewrite Equation (Equation 7) as
(9)$$T\left(E\right)=\frac{\Gamma^{2}{\left(G_{1N}^{0}\right)}^{2}}{{\left|{\left[1+\frac{i\Gamma }{2}G_{NN}^{0}\right]}^{2}+\frac{\Gamma^{2}}{4}{\left(G_{1N}^{0}\right)}^{2}\right|}^{2}}\;.$$

Considering the passage of the electron through the system in a coherent transport, we determine the current passing through the quantum systems. This can be considered as a scattering process of an electron between the contacts. Using the Landauer’s formalism, the I-V characteristics can be obtained by the following expression [27,32]:(10)$$I\left(V\right)=I_{0}\int_{-\infty }^{\infty }\left(f_{L}-f_{R}\right)T\left(E\right)dE\;,$$
where $I_{0}=e/\pi \hbar$ and $f_{L(R)}$ is the Fermi–Dirac distribution function given by $f_{L(R)}=f\left(E-\mu_{L(R)}\right)$, where $\mu_{L(R)}=E_{f}\pm \mathrm{eV}/2$ is the chemical potential.

The noise power of current fluctuations (NPCF) is calculated by the following expression [33,34,35]:(11)$$S=S_{0}\int_{-\infty }^{\infty }T\left(E\right)\left[f_{L}\left(1-f_{L}\right)+f_{R}\left(1-f_{R}\right)+\left\{\left(1-T\left(E\right)\right){\left(f_{L}-f_{R}\right)}^{2}\right\}\right]dE\;,$$
with $S_{0}=2e^{2}/\left(\pi \hbar \right)$. The first two terms of Equation (Equation 11) correspond to the equilibrium noise contribution and the last one gives the nonequilibrium or shot noise contribution to the power spectrum. By calculating the NPCF (*S*) and the total current flowing through the aromatic molecules, the Fano factor can be evaluated via the following relationship [33]:(12)$$F=\frac{S}{2eI}\;.$$

When F=1, the shot noise achieves the Poisson limit for which there is no correlation between the charge carriers. On the other hand, when F<1, the shot noise achieves the sub-Poisson limit and it provides the information about the quantum correlation of the charge carriers [36,37].

To complete the full landscape of electrical conduction in these systems, we evaluate the G-electrical conductance and the thermoelectric properties, such as Q-Seebeck coefficient, κ-thermal conductance, and the ZT-figure of merit [38,39,40,41]. By using Landauer integrals, Li, the following relations are obtained:
(13a)$$G=\frac{2e^{2}}{h}{\mathrm{Li}}_{0}\;,$$
(13b)$$Q=-\frac{1}{e\Theta }\frac{{\mathrm{Li}}_{1}}{{\mathrm{Li}}_{0}}\;,$$
(13c)$$\kappa =\frac{2}{h\Theta }\left({\mathrm{Li}}_{2}-\frac{{\mathrm{Li}}_{1}^{2}}{{\mathrm{Li}}_{0}}\right)\;,$$
and
(13d)$$ZT=\frac{GQ^{2}\Theta }{\kappa }=\frac{1}{\frac{{\mathrm{Li}}_{0}{\mathrm{Li}}_{2}}{{\mathrm{Li}}_{1}^{2}}-1}\;,$$
where Θ is the equilibrium temperature, *e* is the electronic charge, and *h* represents the Plank’s constant. The Landauer integrals used in Equation (13) have the form
(14)$${\mathrm{Li}}_{n}=-\int T\left(E\right){\left(E-E_{f}\right)}^{n}\left(\frac{\partial f(E)}{\partial E}\right)dE\;,$$
where $E_{f}$ describes the equilibrium Fermi energy of the system under zero biased condition, and f(E) gives the Fermi–Dirac distribution function.

The next section describes the analytical and numerical results obtained from the application of the decimation method and Landauer’s formalism for molecular systems connected to terminal electrodes.

## 4. Results and Discussion

### 4.1. Flat Molecule

The calculation of the T(E)-transmission probability of the electron through the molecular system is essential if we want to calculate other transport properties such as the current, current fluctuations, electrical conductance, thermal conductance, and Seebeck coefficient, among others. For this reason, we have considered variations in the coupling between the contacts and the molecular system (Γ), taking the energy value of the carbon as $E_{c}=1.0\;$eV and the corresponding energy bonding values between atoms $t_{\delta }$ and $t_{\vartheta }$ with the same value ($t_{\delta }=t_{\vartheta }$). We use these values to calculate the electrical properties where the molecule is considered flat.

To begin, we validate the renormalization method used in our calculations and determine the transmission probability T(E), where the *N* resonances shown in the transmission profile (Figure 3) coincide with the eigenvalues of the TBH ($H_{0}$) that characterizes the 2,7-Di([1,1′-biphenyl]-4-yl)-9H-fluorene molecular system.

Such Hamiltonian given by Equation (Equation 2) can be represented by the matrix given by
(15)$$H_{0}=\left(\begin{array}{ccccccc} E_{c} & t_{\delta } & 0 & 0 & 0 & t_{\delta } & \ldots \\ t_{\delta } & E_{c} & t_{\delta } & 0 & 0 & 0 & \ldots \\ \vdots & \ddots & \ddots & \ddots & \ddots & \ddots & \ldots \\ \vdots & \vdots & \ddots & \ddots & \ddots & \ddots & \ldots \\ \vdots & \vdots & \vdots & \ddots & \ddots & \ddots & \ldots \\ 0 & 0 & 0 & 0 & t_{\delta } & E_{c} & t_{\delta }\\ 0 & t_{\delta } & 0 & 0 & 0 & t_{\delta } & E_{c} \end{array}\right)\;.$$

The eigenvalues $H_{0}$ have been calculated, finding fourteen positive eigenvalues ([2.42, 2.34, 2.27, 2.12, 1.98, 1.76, 1.57, 1.42, 1.36, 1.12, 0.84, 0.66, 0.42, 0.2] eV), six positive degenerate eigenvalues with values of 1.0eV, thirteen negative eigenvalues ([−0.54,−0.66,−0.9,−1.18,−1.27,−1.52,−1.58,−1.84,−1.95,−2.15,−2.23,−2.35,−2.4] eV), and four negative degenerate eigenvalues with values of −1.0eV. As we can see, these eigenvalues are within the range of the resonances in the transmission probability profile in a weak coupling regime ($t_{\delta },t_{\vartheta }\gg \Gamma ∼0.05\;$eV). This fact validates the renormalization or decimation method applied to the molecule. Once our method has been validated, we proceed to calculate the other thermal and electrical transport properties that are explained below.

In Figure 3, the transmission probability is calculated as a function of the electron injection energy (*E*) and the molecule-contact bonding energy (Γ). Figure 3a represents the transmission probability in an energy range Γ between [0.0−2.0]eV, and Figure 3b is a zoom for values of Γ between [0.0−0.05]eV. We can see that as Γ increases, the bandwidth increases. This phenomenon is related to the fact that the eigenstates of metal contacts hybridize with the eigenstates of the molecular ends. On the other hand, Figure 3a shows two band regions, the first is the region between −2.5 eV < *E* < −0.55 eV and the other one is for values between 0.2 eV < *E* < 2.5 eV. The highest occupied molecular orbital (HOMO) and the lowest unoccupied molecular orbital (LUMO) are located on the edges of these zones and the band gap. The bandwidth of the gap is ∼2.0 eV, which is in good accordance with the work of Na Xin et al. [42].

In Figure 4, the current (*I*), the shot noise (*S*), and the Fano factor (*F*) are calculated as functions of the coupling Γ and the bias voltage (*V*). The current flow occurs due to an imbalance between the chemical potentials of the electrodes (in equilibrium conditions of the chemical potentials, there is no current flow through the molecular system) which depends on the weak or strong coupling that exists with the molecular system. As we can see in Figure 4a, an increase in current occurs with increasing the Γ-coupling, at the same time, the voltage gap where the current is zero goes from ∼2.0 V (where Γ∼0.1eV) to ∼0.6V (where Γ∼2.5eV). The system current, Figure 4a, has a maximum around 1μA (where the current is saturated) for Γ∼0.5eV. This order of magnitude in the current amplitude agrees with the results obtained by Na Xin et al. [42] for the flat molecular system at room temperature. On the other hand, the steps presented in the current profile are due to the resonances that occur in the amplitude of T(E); in other words, when the electrochemical potential coincides with a molecule eigenvalue. The quantum current fluctuations (S) are determined by using Equation (Equation 11). Figure 4b shows the (normalized) current fluctuations $\left(S/S_{0}\right)$ as a function of the bias voltage and the lead-molecule coupling (Γ). The same coupling energy values used in the current have been taken. We can see that by increasing the bias voltage (*V*), for any value of Γ>0, an increase in current fluctuations is presented, giving a growth similar to the characteristic curve of electrical current profile (like the ladder profile), where each step corresponds to an molecule eigenvalue. On the other hand, it is observed that the strong or weak coupling affects the shot noise amplitude. The maximum values of the current fluctuations occur when $\Gamma \geq t_{\delta },t_{\vartheta }$ (strong coupling) and decrease when the coupling between the molecule and the contacts is very small. In Figure 4c, it is observed that for small bias voltages and for any value of Γ, the electronic correlation of the system is null (F=1). In this case, the quantum noise reaches the Poisson limit. After the threshold voltage, which is determined by the energy gap, the quantum noise reaches the sub-Poisson limit (F<1) and provides information on the quantum correlation of the load carriers. In addition to the electrical transport properties already calculated, the thermal properties of transport of charge will be analyzed below.

The Figure 5 shows G-electrical conductance, κ-thermal conductance, Q-Seebeck coefficient, and ZT as a function of energy and temperature (Θ) for the flat 2,7-Di([1,1′-biphenyl]-4-yl)-9H-fluorene molecular system. The same values of the previous calculations have been taken into account, except that a constant value of Γ=0.5eV is used. As we know, the G-electrical conductance (Figure 5a) is proportional to the transmission probability (see Figure 3); therefore, the profiles are similar, i.e., the constructive resonances or peaks presented in G can also represent the molecule eigenvalues. We can also observe that a band-gap is generated due to destructive interference between the localized states of the molecule and the delocalized states of the metal contacts when they are coupled. It is important to highlight that the conductance was calculated within a strong regime (Γ=0.5eV), in such a way that the states of both the molecule and lead hybridize. On the other hand, we can observe that as the temperature increases the G-amplitude decreases. This is because the atoms gain kinetic energy due to the increase in temperature and therefore the amplitude of the vibration of the atoms of the molecular system around their equilibrium positions increases. The molecular vibration within the molecules at their equilibrium behaves like a harmonic oscillator. Thus, an increase in temperature produces an increment at the amplitude on the oscillation between the atoms. Consequently, the movement of the free electrons going from one electrode to the other is hampered and the resistance increases [43]. It increases the interference of the atomic nuclei with the trajectories of valence electrons along the molecule, responsible for the electric conduction, increasing the resistance of the system.

Complementary to the above, we can observe that the width of the energy bands and the antiresonance or band-gap in the κ-thermal conductance are similar (Figure 5b) to those presented in the G-profile. However, in contrast to G, κ increases with temperature; this is an expected result, since the increase in thermal conductance with temperature implies a heat transfer associated with the vibrations of the atoms. The similarity between electrical conductance and thermal conductance is based on the fact that in both, free electrons that pass through the molecule are involved, however, the thermal conductance increases with the average speed of the particles. This fact is because these charges increase the energy transport, while the electrical conductance decreases with the increase in speed of the particles, generating multiple collisions and the deviation of the path of the electrons from one lead to the other. In Figure 5c, it is observed that for the energy values where the band-gap is found, the Q-Seebeck coefficient varies considerably and can reach a zero value at a maximum or minimum of G. We can also observe that Q reaches a finite value that depends on temperature, however, it can increase in magnitude and change its sign as the chemical potential passes through an antiresonance in G[44]. The zeros in the Seebeck coefficient correspond to the minimum values of the ZT-merit figure (Figure 5d), and we also see that Q and ZT increase strongly in the proximity of the destructive interference (band-gap). The existence of nodes, or destructive quantum interference at the transmission probability spectrum, is a characteristic outcome of coherent transport at the system. However, the incoherent processes could also lead to very low transmission probabilities, thus, their effects could be indistinguishable at the electric conductance from the one caused by a transmission node. Although the thermoelectric effects dramatically improve near those transmission nodes, the entropy flux (a quantity inherently incoherent) is not blocked by the destructive quantum interference. Therefore, the existence of nodes give information for coherent transport, being comparable with the experiments at the molecule-electrode junctions. This is due to the fact that the transport of entropy (or disorder’s flow, which depends on Q and ZT) has very little sensitivity to quantum interference and is related to the purity of the molecular system state. A pure system has an entropy current equal to zero, therefore, the existence of a disorder flow implies incoherence—contrary to having electric current which can be completely coherent and can be overridden by destructive quantum interference. Finally, we observe an important behavior when a high degree of ZT appears around the edges of the allowed bands (in G-electrical conductance), which can even reach more than 20 at Θ∼400K; this fact only depends on the asymmetric nature of the conductance on such edges.

### 4.2. Variation of the Two Dihedral Angles between the Two Extreme Benzene Rings

In this section, we analyze the electrical and thermal properties of the 2,7-Di([1,1′-biphenyl]- 4-yl)-9H-fluorene molecule, taking into account the variation of the dihedral angle between the two extreme benzene rings of the molecule. We only take into account a fixed value for Γ=0.5eV, where the current amplitude reached its maximum value. Likewise, it is clear that both the energy parameters of the atomic sites and the couplings are the same as those taken for the plane molecule, except for the $t_{\vartheta }$ coupling, which varies depending on the dihedral rotation angle (such that $t_{\vartheta }=t_{\delta }\;\cos\gamma$), presenting stereoelectronic effects characterized by having an effect on the structure, reactivity, or other properties of the molecule which are generated by the spatial position of the molecule orbitals. Especially, in this case, where the molecular structure is based on biphenyl aromatic components, the stereoelectronic effect implies a variation of the torsion angles and therefore the superposition of the π orbitals of the phenyl rings. Consequently, the properties and performance of devices that can be made from them can be affected [25,42,45,46].

Two cases are presented: the first, when the left dihedral angle (γ) remains fixed—also called strongly-conjugated—and the right rotates a certain angle, also called weakly-conjugated (Figure 6a); and the second one, when the left dihedral angle (γ) varies (weakly-conjugated) and the right remains constant (strongly-conjugated) (Figure 6b). Calculations of the current as a function of voltage bias and of the right dihedral angle rotation, as well as the current as a function of the voltage bias and of the left dihedral angle rotation are shown in Figure 6c,d, respectively. As can be seen, the angle’s variation (γ) is made from 0 to $180^{{}^{\circ}}$. In the first case, we observe that when the right angle increases, the current decreases until it reaches its minimum value or zero at $90^{{}^{\circ}}$, then as it continues rotating, the amplitude of the current increases again reaching its maximum value at $180^{{}^{\circ}}$. This is due to the fact that when there is a maximum current, the states or π orbitals of the two extreme rings of the molecule are strongly conjugated, contrary to the case when these states are weakly conjugated and the molecular system does not conduct electrons. On the other hand, if we vary the left dihedral angle in the same direction that the right one was varied, we find a similar behavior to the situation when the right angle was rotated (see Figure 6b). The foregoing indicates that, depending on the structural variation of the right or left extreme dihedral angles of the 2,7-Di([1,1′-biphenyl]-4-yl)-9H-fluorene molecule, it gives rise to a characteristic behavior of a molecular switch, which can be manipulated by varying any of the dihedral angles of the system; this fact is in agreement with Reference [42]. In the same way, it could be deduced that both the *S*-quantum noise and the *F*-Fano factor behave similar to the flat molecule, except for the fact that the quantum noise decreases or increases when there is a change in the molecular structure and goes from a Poisson limit (F=1 on the weakly conjugated states) to a sub-Poisson limit (F<1 on the strongly conjugated states).

Finally, it can also be inferred that when determining the thermal properties G and κ, depending on the molecular structure or variation of any of the dihedral angles, these properties decrease (compared to the same properties in the flat molecule, Figure 5a,b in the regime of weakly conjugated states (as we can see in Figure 7a,b, respectively), where in this case the left dihedral angle has been rotated $30^{{}^{\circ}}$. At the same time, we can observe that if the electrical conductance (G) decreases with the rotation of any of the dihedral angles, the amplitude of both the Seebeck coefficient (Q) and the ZT increase [41].

## 5. Conclusions

The electrical properties such as transmission probability, I−V characteristic curves, electrical conductance, shot noise, and Fano factor, as well as the thermal properties such as thermal conductance, Seebeck coefficient, ZT, and consequently the stereoelectronic effect, have been studied through the 2,7-Di([1,1′-biphenyl]-4-yl)-9H-fluorene molecular system using the renormalization method in the Hamiltonian framework of tight-binding and Green’s function techniques. These properties were determined, taking into account the variation of the contact-molecule coupling, temperature, and the dihedral angles between the π orbitals of the phenyl rings at the extreme of the molecule. In order to validate the renormalization method, the Hamiltonian eigenvalues of the molecular system were determined, finding a great agreement with the energy values where the resonances were presented in the transmission probability profile in a weak coupling regime. A great agreement was also found in the value of the gap between the HOMO and LUMO bands determined by Xin et al. [19] and our result.

It was demonstrated that the analytical renormalization treatment of the system has presented a reliable approach, giving results that allowed understanding phenomena such as the stereoelectronic effect of molecular conductance. It was also shown that the change in charge transport through the molecule depends on the temperature, the Γ-coupling, and the torsion of the phenyl ring based on a strong conjugation, or high conductance, and on a low conjugation between states, or low electric conductance. This change or commutation of the charge transport induced by the stereoelectronic effect generates high prospects in the construction of molecular devices based on organic materials such as the 2,7-Di([1,1′-biphenyl]-4-yl)-9H-fluorene molecule presented here and can serve as a prototype for applications in molecular electronics.

## Figures and Tables

**Figure 1 molecules-25-03215-f001:**
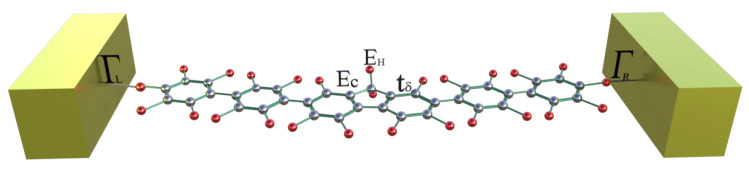
(Color online) Flat 2,7-Di([1,1′-biphenyl]-4-yl)-9H-fluorene molecular aromatic system.

**Figure 2 molecules-25-03215-f002:**
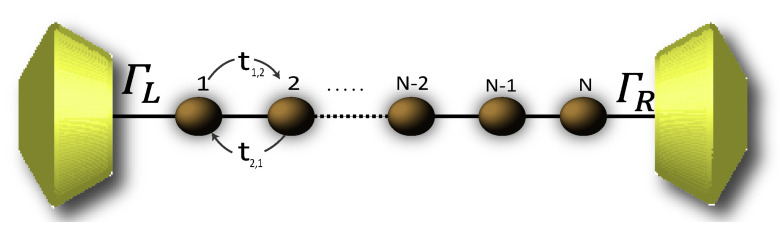
(Color online) Representation of the effective molecule coupled with the contacts.

**Figure 3 molecules-25-03215-f003:**
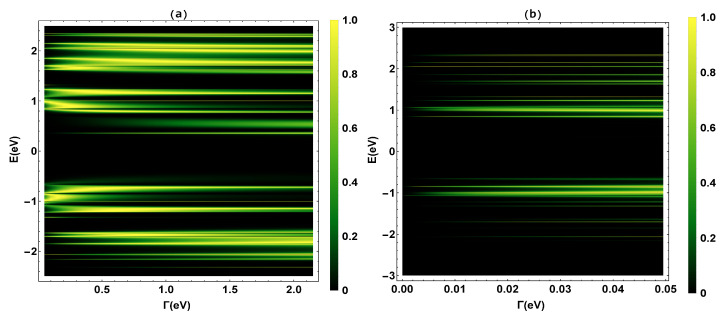
(**a**) Transmission probability as a function of the molecule–electrode coupling (Γ) and the energy (*E*). (**b**) A zoom of the transmission probability for 0.0005eV<Γ<0.05eV.

**Figure 4 molecules-25-03215-f004:**
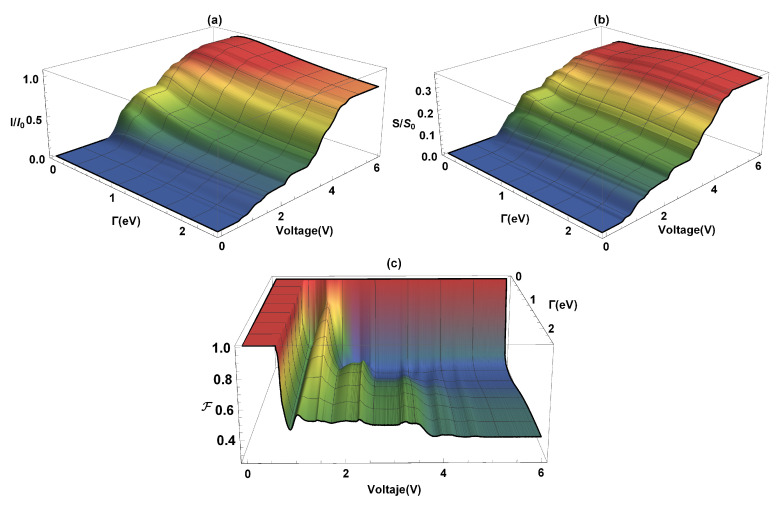
(**a**) Current, (**b**) current fluctuations, and (**c**) Fano factor as a function of the bias voltage (*V*) and the coupling lead-molecules (Γ).

**Figure 5 molecules-25-03215-f005:**
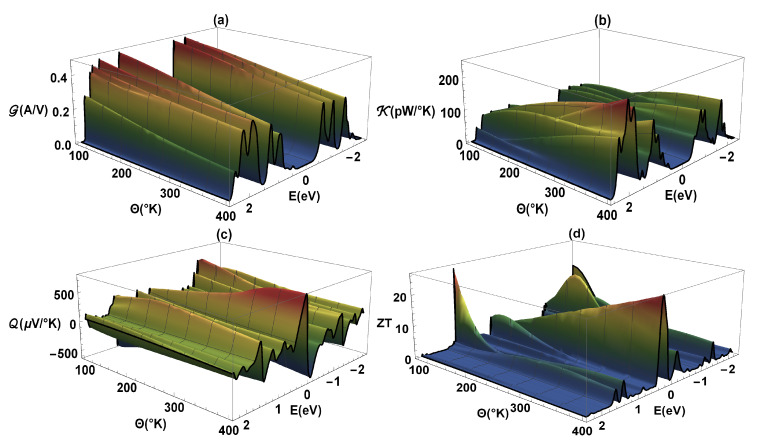
(**a**) Electrical conductance (G), (**b**) thermal conductance (κ), (**c**) Seebeck coefficient (Q), and (**d**) ZT as a function of energy and temperature (Θ).

**Figure 6 molecules-25-03215-f006:**
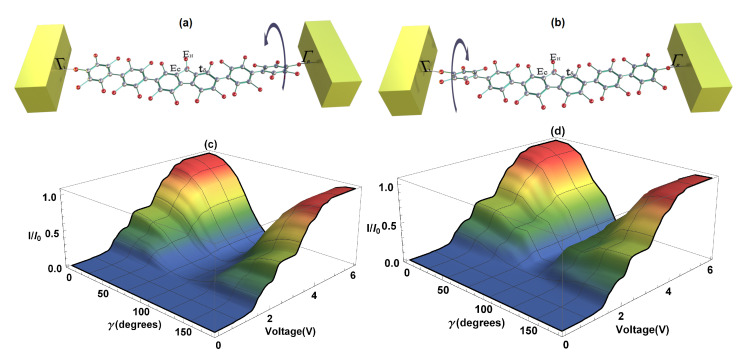
(**a**) Representation of the right dihedral angle rotation, (**b**) representation of the left dihedral angle rotation, (**c**) current as a function of bias voltage and of the right dihedral angle rotation, (**d**) current as a function of bias voltage and of the left dihedral angle rotation.

**Figure 7 molecules-25-03215-f007:**
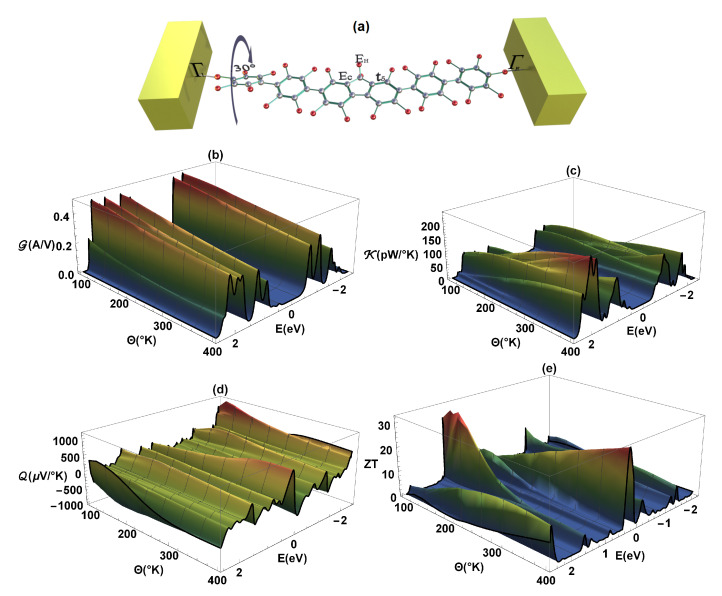
(**a**) Representation of the left dihedral angle rotation ($30^{{}^{\circ}}$), (**b**) electrical conductance (G), (**c**) thermal conductance (κ), (**d**) Seebeck coefficient (Q), and (**e**) ZT as a function of energy and temperature (Θ).

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
