# Peer review of "Thermo-Electrical Conduction of the 2,7-Di([1,1′-Biphenyl]-4-yl)-9H-Fluorene Molecular System: Coupling between Benzene Rings and Stereoelectronic Effects"

_molecules, 2020, doi:10.3390/molecules25143215_

Round 1

Reviewer 1 Report

In this manuscript, thermal and electrical properties of a fluorene derivatives were theoretically studied.

This should be a kind of article having a quite limited interest for particular researchers, and the specific molecule of fluorene derivative is restrictively treated in detail. Therefore, the reviewer does not have any positive comments for this article.

The authors should examine the generallizability of the results. As quite expected, the difedral angles between benzene rings affect the properties,  which are low novelty. 

Author Response

We would like to thank the reviewer for his/her recommendations and views.

After reviewing the Referee's comments and critically analyzing our manuscript, we believe that the Referee is absolutely right. Our presentation of the study system and methodology was not shown properly. We were not careful to present the news of our work. We do not adequately show the contributions that our research makes to the study system. We were not careful in showing the power of the method used. We were not careful to show the technological relevance of the studied molecule. In this revised version of the manuscript, we have tried to correct these aspects. We deeply appreciate the Referee because his/her opinion motivated us to substantially improve the quality of our work. We believe that our research may be of interest to a very broad community of researchers on the subject of molecular systems and the methods for interpreting the physics-chemistry involved.

The introduction was complemented giving a broader to the scientific community, especially in the area of molecular electronics, a greater understanding. On the other hand, to complement the description, design, method, results, and conclusions of the research, the following paragraphs were added:

On page 2 of the revised version of the manuscript:

“Over the past few years, researchers in chemistry, physics, and nanotechnology have begun to understand stereoelectronics adequately. This is the means through which it is possible to interpret how and why the electronic properties of molecules vary and can be modified in the process of shaping physical-chemical systems. With the appearance of single-molecule techniques, the necessary instruments and technology have been created to follow the conformation of molecules step by step. By introducing the stereoscopic effect on, for example, in the aromatic structure of biphenyl, it is possible to modify the structural and physical properties of an organic functional molecule.  Aiming to investigate the process of intramolecular conformation of biphenyl and how this affects the dynamics of its transport properties at the level of an isolated molecule, Xin et al. [19] designed an aromatic hexaphenyl chain molecule. The medial biphenyl was placed using a form of fluorene in order to fix the dihedral angle. Additionally, it has been covalently sandwiched between the contacts of a graphene dot, and thus, stable single-molecule bonds have been created. The authors report, mainly at a temperature of 120 K, a stochastic change between two different states of conductance in the I-V curves, and have demonstrated that they remain in a state of low conductance up to 140 K. Biphenyl, as the elementary unit of organic functional materials, has been shown to have significant potential for use in organic field-effect transistors, organic light-emitting diodes, and solar cells. Both experimental and theoretical researchers suggest that the stereoelectronic effect of the biphenyl-based aromatic structures can significantly affect the structural and photophysical properties of organic functional molecules [20-25]. Li et al. [26] investigated stereoelectronic effects in a hexaphenyl aromatic chain. In their work, the central biphenyl was introduced in a fluorene form to fix the dihedral angle. Following the same approach, and in order to analyze the single-molecule charge transport properties, the authors have included an amine-terminated hexaphenyl aromatic chain located between carboxylic-acid-terminated graphene electrodes.”

On page 3 of the revised version of the manuscript:

“The renormalization method is a method that transforms the real space into an effective space that generates the systematic reduction of the Hamiltonian dimension, i.e., the set of linear equations that characterize the system becomes a set of nonlinear equations with effective energies and couplings with fewer degrees of freedom. The renormalization method is a highly efficient and very low computational cost (compared to ab-initio, such as density functional theory (DFT) methods). For this reason, to calculate the transport properties in this work, the renormalization process is done under the inclusion of Feynman paths using the Green functions, which can be considered as propagators (Dyson's equation) that carry the charge transport information through the molecular system and provide an alternative in the solution of the Schrödinger equation in a numerical and analytically stable way. It is important to note that the renormalization method used in this work has been implemented in the calculation of quantum transport properties through quantum wire systems, aromatic molecular systems, DNA molecules, among others, and have been compared with transport properties through similar systems, determined by DFT calculations and/or experimental results [28-31,35].”

On page 7 of the revised version of the manuscript:

“The molecular vibration within the molecules at their equilibrium behaves like a harmonic oscillator. Thus, an increase in temperature produces an increment at the amplitude on the oscillation between the atoms.”

On page 8 of the revised version of the manuscript:

“It increases the interference of the atomic nuclei with the trajectories of valence electrons along the molecule, responsible for the electric conduction, increasing the resistance of the system.”

On page 9 of the revised version of the manuscript:

“The existence of nodes, or destructive quantum interferences at the transmission probability spectrum, is a characteristic outcome of coherent transport at the system. However, the incoherent processes could lead as well to very low transmission probabilities, thus, their effects could be undistinguishable at the electric conductance from the one caused by a transmission node. Although the thermoelectric effects dramatically improve near those transmission nodes, the entropy flux (a quantity inherently incoherent) is not blocked by the destructive quantum interference. Therefore, the existence of nodes gives information for coherent transport, being comparable with the experiments at the molecule-electrode junctions.”

On the conclusions section of the revised version of the manuscript:

“In order to validate the renormalization method, the Hamiltonian eigenvalues of the molecular system were determined, finding a great agreement with the energy values where the resonances were presented in the transmission probability profile in a weak coupling regime. Such as a great agreement was also found in the value of the gap between the HOMO and LUMO bands determined by Xin et al. [19] and our result.”

Subsequently, we have reduced the appendix and added the following useful information: ….(with N=12)…

The grammar and redaction of the manuscript were revisited in order to improve the quality of the text.

Finally, the references (1, 20, 21, 22, 23, 24, 25 y 26) were added in the revised version of the manuscript.

Reviewer 2 Report

See comments to the editor

I had to choose accept to be able to send this form but I'm just declining the ability to review the paper

Author Response

No comments

Reviewer 3 Report

I think this manuscript would be appropriate for publication in Molecules only after simplification for general auditory (for peoples who not very familiar in mathematical aspects). The methodology of research is not routine. The results are presented and discussed briefly and not very clear. Probably, this manuscript would be interesting for readers with strong mathematical background, but not for pure chemists.

The manuscript interesting to read and well illustrated. The introduction section gives some background into the topic of author's research. The references list is appropriate. The level of English language is quite understandable for me (however, I am not a native speaker, and, probably, some polishing of several phrases by the editorial team could be useful). Overall, I think this manuscript would be appropriate for publication in Molecules only after simplification for general auditory (for peoples who not very familiar in mathematical aspects). The methodology of research is not routine. The results are presented and discussed briefly and not very clear. Probably, this manuscript would be interesting for readers with strong mathematical background, but not for pure chemists. I suggest to the authors to revise and rewrite this manuscript in a simpler and more detailed style as clear as possible for general audience of the Molecules journal.

Author Response

We would like to thank the reviewer for his/her recommendations and views.

The introduction was complemented giving a broader to the scientific community, especially in the area of molecular electronics, a greater understanding. On the other hand, to complement the description, design, method, results, and conclusions of the research, the following paragraphs were added:

On page 2 of the revised version of the manuscript:

“Over the past few years, researchers in chemistry, physics, and nanotechnology have begun to understand stereoelectronics adequately. This is the means through which it is possible to interpret how and why the electronic properties of molecules vary and can be modified in the process of shaping physical-chemical systems. With the appearance of single-molecule techniques, the necessary instruments and technology have been created to follow the conformation of molecules step by step. By introducing the stereoscopic effect on, for example, in the aromatic structure of biphenyl, it is possible to modify the structural and physical properties of an organic functional molecule.  Aiming to investigate the process of intramolecular conformation of biphenyl and how this affects the dynamics of its transport properties at the level of an isolated molecule, Xin et al. [19] designed an aromatic hexaphenyl chain molecule. The medial biphenyl was placed using a form of fluorene in order to fix the dihedral angle. Additionally, it has been covalently sandwiched between the contacts of a graphene dot, and thus, stable single-molecule bonds have been created. The authors report, mainly at a temperature of 120 K, a stochastic change between two different states of conductance in the I-V curves, and have demonstrated that they remain in a state of low conductance up to 140 K. Biphenyl, as the elementary unit of organic functional materials, has been shown to have significant potential for use in organic field-effect transistors, organic light-emitting diodes, and solar cells. Both experimental and theoretical researchers suggest that the stereoelectronic effect of the biphenyl-based aromatic structures can significantly affect the structural and photophysical properties of organic functional molecules [20-25]. Li et al. [26] investigated stereoelectronic effects in a hexaphenyl aromatic chain. In their work, the central biphenyl was introduced in a fluorene form to fix the dihedral angle. Following the same approach, and in order to analyze the single-molecule charge transport properties, the authors have included an amine-terminated hexaphenyl aromatic chain located between carboxylic-acid-terminated graphene electrodes.”

On page 3 of the revised version of the manuscript:

“The renormalization method is a method that transforms the real space into an effective space that generates the systematic reduction of the Hamiltonian dimension, i.e., the set of linear equations that characterize the system becomes a set of nonlinear equations with effective energies and couplings with fewer degrees of freedom. The renormalization method is a highly efficient and very low computational cost (compared to ab-initio, such as density functional theory (DFT) methods). For this reason, to calculate the transport properties in this work, the renormalization process is done under the inclusion of Feynman paths using the Green functions, which can be considered as propagators (Dyson's equation) that carry the charge transport information through the molecular system and provide an alternative in the solution of the Schrödinger equation in a numerical and analytically stable way. It is important to note that the renormalization method used in this work has been implemented in the calculation of quantum transport properties through quantum wire systems, aromatic molecular systems, DNA molecules, among others, and have been compared with transport properties through similar systems, determined by DFT calculations and/or experimental results [28-31,35].”

On page 7 of the revised version of the manuscript:

“The molecular vibration within the molecules at their equilibrium behaves like a harmonic oscillator. Thus, an increase in temperature produces an increment at the amplitude on the oscillation between the atoms.”

On page 8 of the revised version of the manuscript:

“It increases the interference of the atomic nuclei with the trajectories of valence electrons along the molecule, responsible for the electric conduction, increasing the resistance of the system.”

On page 9 of the revised version of the manuscript:

“The existence of nodes, or destructive quantum interferences at the transmission probability spectrum, is a characteristic outcome of coherent transport at the system. However, the incoherent processes could lead as well to very low transmission probabilities, thus, their effects could be undistinguishable at the electric conductance from the one caused by a transmission node. Although the thermoelectric effects dramatically improve near those transmission nodes, the entropy flux (a quantity inherently incoherent) is not blocked by the destructive quantum interference. Therefore, the existence of nodes gives information for coherent transport, being comparable with the experiments at the molecule-electrode junctions.”

On the conclusions of the revised version of the manuscript:

“In order to validate the renormalization method, the Hamiltonian eigenvalues of the molecular system were determined, finding a great agreement with the energy values where the resonances were presented in the transmission probability profile in a weak coupling regime. Such as a great agreement was also found in the value of the gap between the HOMO and LUMO bands determined by Xin et al. [19] and our result.”

Subsequently, we have reduced the appendix and added the following relevant: ….(with N=12)…

The grammar and redaction of the manuscript were revisited in order to improve the quality of the text.

Finally, the references (1, 20, 21, 22, 23, 24, 25 y 26) were added in the revised version of the manuscript

Round 2

Reviewer 1 Report

Thank you for your sincerely revision. 

I would like to recommend acceptance of your manuscript.